# Relative and Absolute Quantification of Aberrant and Normal Splice Variants in *HBB^IVSI−110 (G > A)^* β-Thalassemia

**DOI:** 10.3390/ijms21186671

**Published:** 2020-09-11

**Authors:** Petros Patsali, Panayiota Papasavva, Soteroulla Christou, Maria Sitarou, Michael N. Antoniou, Carsten W. Lederer, Marina Kleanthous

**Affiliations:** 1Department of Molecular Genetics Thalassaemia, The Cyprus Institute of Neurology and Genetics, Nicosia 1683, Cyprus; petrospa@cing.ac.cy (P.P.); panayiotap@cing.ac.cy (P.P.); marinakl@cing.ac.cy (M.K.); 2Cyprus School of Molecular Medicine, Nicosia 1683, Cyprus; 3Thalassaemia Clinic Nicosia, Ministry of Health, Nicosia 1474, Cyprus; chrnchr@spidernet.com.cy; 4Thalassaemia Clinic Larnaca, Ministry of Health, Larnaca 6301, Cyprus; msitarou@yahoo.gr; 5Department of Medical and Molecular Genetics, King’s College London, London SE1 9RT, UK; michael.antoniou@kcl.ac.uk

**Keywords:** β-thalassemia, splice defect, duplex quantitative PCR, absolute quantification, transcript variants, splicing

## Abstract

The β-thalassemias are an increasing challenge to health systems worldwide, caused by absent or reduced β-globin (HBB) production. Of particular frequency in many Western countries is *HBB^IVSI−110(G > A)^* β-thalassemia (HGVS name: HBB:c.93-21G > A). Its underlying mutation creates an abnormal splice acceptor site in the *HBB* gene, and while partially retaining normal splicing of *HBB*, it severely reduces HBB protein expression from the mutant locus and *HBB* loci in trans. For the assessment of the underlying mechanisms and of therapies targeting β-thalassemia, accurate quantification of aberrant and normal *HBB* mRNA is essential, but to date, has only been performed by approximate methods. To address this shortcoming, we have developed an accurate, duplex reverse-transcription quantitative PCR assay for the assessment of the ratio and absolute quantities of normal and aberrant mRNA species as a tool for basic and translational research of *HBB^IVSI−110(G > A)^* β-thalassemia. The method was employed here to determine mRNA ratios and quantities in blood and primary cell culture samples and correlate them with HBB protein levels. Moreover, with its immediate utility for β-thalassemia and the mutation in hand, the approach can readily be adopted for analysis of alternative splicing or for quantitative assays of any disease-causing mutation that interferes with normal splicing.

## 1. Introduction

The *HBB^IVSI−110(G > A)^* β-thalassemia mutation (*HBB^IVSI−110^*) is classified as a Mediterranean mutation, but through migration has spread globally, with peak relative carrier frequencies in endemic countries, such as Cyprus (76%), Greece (42%), Turkey and North Macedonia (both 38%) and Egypt (37%), and lower frequencies in target countries of mixed migration, such as Germany (25%) and the United Kingdom (5%) (source: ITHANET [1]). The mutation introduces an aberrant splice site in the *HBB* gene, leading to integration of 19 nucleotides of intron 1, including an in-frame stop codon, into the corresponding abnormal mRNA. The latter is thus assumed to be an efficient target for nonsense-mediated decay [2,3]. Based on the studies performed to date, it is still unknown how the absolute quantities of total *HBB* mRNA compare in patients and controls. However, for relative quantities of normal to aberrant HBB mRNA and based on semi-quantitative RT-PCR assessment, it appears that in reticulocytes and peripheral blood of patients homozygous for the *HBB^IVSI−110^* mutation, normal *HBB* mRNA predominates by far [4]. Two other research groups instead investigated liquid cultures of primary hematopoietic cells from different *HBB^IVSI−110^* patients, reporting relative quantities of normal transcripts between 60% and 100% in the one case [5] and of approximately 50% in the other [6,7,8]. However, transgenic expression of *HBB^IVSI−110^* in HeLa (human cervical cancer) and Vero (monkey kidney) cell lines suggested that only up to 20% of steady-state *HBB*-derived mRNA was correctly spliced [9,10]. It is unclear whether these discrepancies were in part brought about by differential nonsense-mediated decay activity or are otherwise inherent to the different biological systems used, which would be in line with apparent differential processing of *HBB* pre-mRNA in different tissues [4]. Given the poor reproducibility of endpoint-PCR-based quantification, which moreover relies on the quantification of electrophoretic band intensities, the quantification method itself additionally reduces comparability between studies and reliable correlation to other disease parameters. For instance, the high relative expression of normal *HBB* mRNA in erythropoietic cells from *HBB^IVSI−110^* homozygotes is apparently at odds with a mere 10% steady-state HBB protein level in those patients compared with that observed in normal individuals. This low protein expression results in transfusion dependence and a severe thalassemia major phenotype for what is ostensibly a mild mutation with residual HBB expression (β^+^ mutation). Moreover, compound heterozygosity for the mutation results in severe thalassemia phenotypes, comparable to the combination with mutations characterized by a complete absence of HBB expression (β^0^ mutations) [11,12]. Data by Breda et al. for *HBB^IVSI−110^* homozygous cells indicated that protein expression from a normal exogenous *HBB* locus, such as that supplied for gene therapy by gene addition, is below that expected for a given vector copy number or HBB mRNA level [7,8]. Besides its epidemiological significance, the *HBB^IVSI−110^* mutation is, therefore, also of special interest owing to its clinical significance for homo- and compound heterozygous patients, for our understanding of protein expression in the presence of aberrant transcripts and for the development of curative therapies.

To quantify the apparent effect of the *HBB^IVSI−110^* locus on the one hand and to allow correlation of *HBB* mRNA and protein levels in the efficiency evaluation of novel therapies on the other, it is necessary to monitor accurately both, steady-state total *HBB*-derived mRNA levels and the ratio of normal to aberrantly spliced RNA. For instance, optimization of measurement would improve the evaluation of the potential of β-thalassemia *HBB^IVSI−110^*-specific gene therapy approaches [13,14,15]. In this study, we have, therefore, developed a duplex reverse-transcription quantitative PCR (RT-qPCR) assay for robust measurement of both mRNA species in patient-derived cells, and have employed it for mRNA measurements and correlation with HBB protein levels for proof of principle.

## 2. Results

### 2.1. Recombinant Plasmids for Standard Curve Creation

We set out to establish a sensitive multiplex RT-qPCR method based on variant-specific probes, a common set of primer pairs and a plasmid clone of normal and aberrantly spliced *HBB* cDNA sequences to generate a standard curve (SC) (Figure 1). This method uses a common pair of primers and two specific probes for detection of the normal and the aberrant mRNA (Figure 1A), respectively, and for accurate quantification relies on a plasmid-based SC as template for amplicons representing both mRNAs. In that respect, providing both templates on separate plasmids (1-insert; 1i) would allow varying same-tube ratios of normal to aberrant template and could eliminate possible interference between both amplicons by allowing single-template reactions. Conversely, providing both templates on the same plasmid (2-insert; 2i) would eliminate pipetting and measurement errors in the determination of relative quantities between both amplicons, while potentially imposing constraints on amplification of templates that are proximal to one another or are held on the same covalently closed circular plasmid DNA. We therefore decided to test split circular 1i simplex, combined circular 1i duplex, and circular, linearized and fragmented 2i template designs for SC creation (Figure 1B).

### 2.2. Evaluation of Alternative Standard Curve Designs

For linearized and fragmented SC designs, plasmid templates were digested as illustrated in Figure 2. Initial evaluation of three different PCR kits, TaqMan Master Mix (Applied Biosystems, Foster City, CA, USA), QuantiTect multiplex kit (Qiagen, Hilden, Germany) and Qiagen Multiplex PCR Kit (Qiagen, Hilden, Germany) showed substantial differences in performance and indicated the latter kit as the most sensitive and reproducible for the analyses in hand (see Appendix A for an exemplary comparison of TaqMan Master Mix with Qiagen Multiplex PCR Kit). Therefore, all subsequent analyses for the current study were performed using the Qiagen Multiplex PCR Kit (Qiagen, Hilden, Germany).

For a quantification based on a split circular SC design (circular 1i simplex SC), four SC dilution points for pCR2.1_HBB_N and pCR2.1_HBB_A in a 1:1 molar ratio were employed. SCs based on the split circular amplicon design gave good sensitivity and efficiency of the reactions, with detection down to 4015 (log_10_ = 3.604) aberrant or normal molecules per reaction (Figure 3A). Notably, peaks for both variant-specific constructs did not fall together (data not shown), contrary to what would be expected at a 1:1 molar ratio of both templates. As both amplicons for this duplex reaction were amplified with the same primer pair and had a size difference of only 15.6% (relative to the smaller 103 nt amplicon), the most likely cause for the observed discrepancy was dilution and spectrophotometric measurement errors. Both types of errors would be eliminated by providing both amplicons with plasmid pCR2.1_HBB_N+A instead, which harbors both amplicons as single copies in tandem in the same orientation. Note, insertion of the two highly similar cDNA *HBB* fragments (normal and aberrant) on the same construct in the same orientation avoids the creation of secondary DNA structures, such as large hairpins, which would be expected to form and to have an impact on the efficiency of the reaction for inverted repeats [16]. Indeed, an SC made up of 10-fold serial dilutions of plasmid pCR2.1_HBB_N+A consistently showed similar threshold cycle (Ct) numbers for both transcript variants at the same data point (Figure 3A). It moreover showed higher sensitivity compared to split curve design, with detection of down to 402 (log_10_ = 2.604) molecules and greater reproducibility between experiments. However, for any of these reactions based on covalently closed circular plasmid templates, the qPCR efficiencies were still lower (with 60.16–67% for normal *HBB* and 49.35–53.01% for aberrant *HBB*) compared to those observed with cDNA samples (with 97.9% and 93.6%, respectively) (Figure 3B). In an attempt to improve the accuracy, reproducibility, sensitivity and efficiency of the method, we decided to evaluate SCs with different structural conformations in order to remove parameters with potential negative impact on the efficiency of the reaction, such as the structure of the constructs (relaxed or supercoiled) that could impair the accessibility of the templates for primers or probes [17]. This was addressed with the linearization of the pCR2.1_HBB_N+A 2-insert construct using the unique SpeI restriction site, which is located between the two *HBB* mRNA fragment variants (linear 2i SC), or with sequential digest with EcoRI/XbaI, which releases both templates as separate fragments from the vector backbone (fragmented 2i SC) (Figure 2 and Figure 3B). Either cleavage strategy, but in particular the use of a fragmented 2i SC, would predictably increase efficiency of the reaction by (i) reducing topological constraints on the DNA, imposed by closely adjacent amplicons, and (ii) reducing or preventing co-amplification of adjacent sequences, such as the intervening insert or the plasmid backbone as part of a combined amplicon. Importantly, prevention of a chimeric amplicon, which would otherwise interfere with detection of either variant, would also increase the accuracy of the standard curve. In line with these considerations, multiplex qPCR efficiency, accuracy and reproducibility for the detected Ct values for each construct in the same dilutions was clearly improved, in particular when the pCR2.1_HBB_N+A fragmented 2i SC was used. Remarkably, use of the fragmented 2i SC greatly increased sensitivity of the reaction and expanded the detection range by 1 log compared to other, circular and linear, 2i SCs, extending reliable detection down to only four (log_10_ = 0.604) molecules per reaction (Figure 3B).

Overall, a multiplex comparison of SCs based on pCR2.1_HBB_N/pCR2.1_HBB_A (1i SCs) or on circular, linear and fragmented pCR2.1_HBB_N+A (2i SCs) clearly established the fragmented 2i SC as superior for PCR efficiency, reproducibility and accuracy of detected Ct values for both the normal and the aberrant amplicons. Efficiency and correlation data are summarized in Table 1.

In order to lower reagent cost for RT-qPCR analyses, reaction volume for the Qiagen Multiplex PCR Kit was halved from the recommended 25 μL to 12.5 μL, by halving all reaction components except a constant 2 μL of cDNA with a corresponding adjustment of the volume of distilled water. The reduction maintained SC linearity and reproducibility for normal (Appendix A) and aberrant (Appendix A) *HBB* mRNA, albeit at lower endpoint fluorescence (Appendix A), prompting our adoption of half-volume reactions for all subsequent RT-qPCR analyses.

### 2.3. mRNA Levels in HBB^IVSI−110^-Homozygous Primary Cells

All *HBB^IVSI−110^*-homozygous patients in Cyprus are transfusion-dependent in order to maintain blood hemoglobin levels above 8 g/dL, in line with national policy for the standard of care for thalassemia patients. Accordingly, analysis of protein and likely of mRNA extracted from peripheral blood of patients would partially reflect protein expression from donor-derived erythrocytes and thus be uninformative. We therefore utilized erythroid cultures and in vitro differentiation procedures for patient-derived CD34^+^ hematopoietic stem and progenitor cells (HSPCs) [18] in order to allow analyses of patient-specific expression.

Applying the established RT-qPCR protocol and utilizing the pCR2.1_HBB_N_A fragmented 2i SC, we measured aberrantly and correctly spliced *HBB* mRNA in cultures of erythroid progenitor cells sampled from normal and *HBB^IVSI−110^* individuals. Across six independent cell culture experiments employing primary cells from four different patients, aberrant transcripts constituted 60.9 ± 7.3% of total *HBB*-encoded mRNA (against 39.1 ± 7.3% of normal transcripts, *p* = 0.0146). Based on similar amplification efficiencies between the fragmented 2i SC and cDNAs from primary cultures (see Figure 3B and Table 1), a comparison of the percentages of correct splicing in primary samples using the 2 ^−(ΔΔCt)^ method and the fragmented 2i SC-based equations gave virtually identical measurements (Pearson correlation coefficient r = 0.925; *p* value 0.001) (Figure 4A, with detail for 2i SC-based measurement in Figure 4B). By contrast, approximate quantification of relative transcript levels based on end-point PCR for the same samples, including a measurement of band intensities and cycle sequencing followed by decomposition of mixed sequence traces [19,20], consistently underestimated the contribution of aberrant *HBB*-derived mRNA (Appendix A). RT-qPCR-based analysis of the levels of all *HBB* variants (wtHBB, IVSI-110 HBB and total HBB) normalized to *HBA* expression levels using the 2 ^−(ΔΔCt)^ method and expressed relative to patient samples, indicated similar normalized steady-state levels of total *HBB* mRNA in cell cultures from normal controls (1.1 × 1.3 ^±1^) and patient samples (1.0 × 2.2 ^±1^). By contrast, normalized levels of the correctly spliced *HBB* mRNA in normal controls (2.7 × 1.4 ^±1^) greatly exceeded that in patient samples (1.0 × 2.1 ^±1^), meaning that cultured cells from patients express 37.2% × 2.1 ^±1^ of normal levels of the functional *HBB* mRNA (Figure 4C).

### 2.4. Correlation of mRNA and Protein Levels in HBB^IVSI−110^ Primary Erythroid Cultures

We then used patient-derived HSPCs after in vitro differentiation for correlation of protein expression with the transcript levels detected in this study. RP-HPLC quantification of all globin chains clearly showed a difference in the expression pattern of all *HBA*-normalized *HBB*-like globin chains between PB samples (*n* = 11) of normal individuals, normal HSPC cultures (*n* = 4) and *HBB^IVSI−110^*-homozygous (*n* = 6) HSPC cultures (Figure 4D). Against PB samples of normal individuals as reference for a 1.09 ± 0.11 HBB/HBA ratio, normal HSPC cultures revealed a 0.87 ± 0.11 HBB/HBA ratio, significantly lower than that of PB (*p* = 0.0038) and indicating differences in globin expression patterns in erythroid cells in culture and PB. Nevertheless, in *HBB^IVSI−110^*-homozygous cultures, globin ratios of 0.20 ± 0.06 for HBB/HBA were significantly lower than either in normal PB or in normal HSPC cultures (*p < 0.0001*) (Figure 4E).

In the following, cultures derived from normal HSPCs were used as reference for relative protein quantification in *HBB^IVSI−110^*-homozygous cultures. On average, for *HBB^IVSI−110^*-homozygous cultures the functional *HBB* mRNA level was 37.2% × 2.06 ^±1^ and the HBB protein level 23.16 ± 6.78% of that observed for normal cultures (mRNA: 100% × 1.44 ^±1^; Protein: 100 ± 20.25%), showing only moderate correlation of mRNA and protein levels for *HBB^IVSI−110^* thalassemia (Pearson correlation coefficient r = 0.494 of functional *HBB* mRNA vs. HBB protein levels, *p = 0.213*) across all samples. Mindful of the possible contribution to this disconnect between *HBB* mRNA and protein level by aberrant globin mRNA with its apparent inhibitory effect on HBB globin production, we conducted a differential analysis of *HBB* transcript variants (normal, aberrant and total *HBB* transcripts) for patient-derived samples, only. For these samples, cultures with relatively high levels of *HBB* transcripts had relatively low *HBB*/*HBA* levels and vice versa (Figure 4F), and a correlation analysis showed a strong correlation between all parameters. Specifically, levels of all *HBB* transcript variants (normal, aberrant and total *HBB* transcript) had a strong positive linear relationship with one another, whereas a comparison of all transcript levels with HBB/HBA chain levels showed a strong negative linear relationship. Although modulation of the fixed ratio of aberrant and normal globin mRNA quantities in *HBB^IVSI−110^*-homozygous cells would be required for a definitive statement, a disproportionate negative effect of aberrant *HBB^IVSI−110^* mRNA on HBB expression for increasing total *HBB*-derived mRNA levels would readily explain the phenomenon.

## 3. Discussion

This study introduces a sensitive and accurate duplex RT-qPCR assay for the quantification of aberrant and normal *HBB^IVSI−110^*-derived mRNA species. We then applied this assay to evaluate for the first time absolute and relative *HBB* mRNA quantities in primary *HBB^IVSI−110^*-homozygous thalassemic cells. Finally, we performed additional protein quantification by RP-HPLC to allow a meaningful correlation of the relative and absolute levels of *HBB* mRNA species and HBB protein levels, providing quantitative evidence that increasing mRNA production from the *HBB^IVSI−110^* locus interferes with HBB protein production.

The newly developed duplex RT-qPCR assay utilizes a plasmid-based fragmented 2i SC, common primers and two transcript-specific probes for aberrant and normal mRNAs produced from the *HBB^IVSI−110^* locus. It is sensitive down to below five template molecules per reaction and of utility for basic and translational studies concerning *HBB^IVSI−110^* patients. In this study, we have shown that utilization for the SC of a single construct holding both templates (2i), which ensures the molar equivalence of both variants in each multiplexing reaction, increased the accuracy of the method compared to SCs based on separate (1i) plasmids in single or duplex reactions. Importantly, we have shown that releasing the *HBB* cDNA fragments from the plasmid backbone (fragmented 2i SC) led to a major improvement of qPCR reaction efficiency and accuracy compared to SCs with circular or linearized 2i constructs, and allowed efficiencies similar to those observed with sample cDNA. DNA confirmation of plasmid-based SC (closed circular/supercoiled, nicked circular and linear) is known to affect qPCR efficiency [17,21,22,23] and will be a major contributing factor to differences between different SCs evaluated here. Supercoiled circular constructs as the majority of purified bacterial plasmids and in contrast to linear cDNA are structurally constraint during the qPCR denaturation step, which interferes with the binding of primers (and probes) and renders them comparably poor DNA templates [22]. Supercoiled DNA is, moreover, sensitive to physical and mechanical stresses common in the laboratory, and the corresponding handling-related introduction of nicks and resultant transition to relaxed circular or linear template molecules increases PCR efficiency while introducing variation in SC-based measurements. In line with these considerations, all circular SCs (circular 1i simplex or duplex (1:1) SCs; circular 2i SC), exhibit lower efficiencies (62–70% normal *HBB*; 57–68% aberrant *HBB*) compared to cDNA serial dilution SC (97%; normal *HBB*; 94%; aberrant *HBB*) in this study. Notably, we observed that the efficiency with the combined (pCR2.1_HBB_N_A) linear 2i SC was even lower (57%; normal *HBB*; 55%; aberrant *HBB*). This phenomenon may possibly be explained by efficient intramolecular hairpin formation and linear amplification of the vector backbone for linear 2i SC constructs, which would interfere with amplification of the SC amplicons, but this contingency was not investigated further. Taken together, we believe that in multiplex reactions containing both templates within the plasmid backbone we observe reduced efficiencies owing to: (a) the formation of chimeric/recombinant amplicons [24] and consequent secondary structures interfering with primer and probe binding [25] and (b) removal of primer binding sites from initial-round amplicons due to 5–3 exonuclease activity of the Hot Start TaqMan Polymerase and the relatively close proximity of the two cDNA amplicons in linear (4073 bp) and especially the circular (247 bp) 2i construct. Both of the suggested inhibitory factors, in addition to supercoiled circular conformation, were effectively minimized by using fragmented 2i SC. In turn, the fragmented 2i SC emerged superior to other SC designs, since it ensures the equal loading of normal and aberrant cDNA fragments as templates, avoids pipetting errors, avoids structural constraints and excludes amplification of any flanking non-cDNA sequences.

In conclusion and prompted by our efficiency data across all SC designs, transcripts from primary erythroid cultures were subsequently either quantified using same-run fragmented 2i SCs or by importing the external single-plasmid SC from a different run for re-assessment by fragmented 2i SC, as established [26]. The latter procedure allowed reanalysis in this study of older RT-qPCR data obtained with a linear 2i SC design [14], without re-running all samples and relied on the inclusion of at least three independent triplicate samples from the original run as points of reference for reliable recalculation of transcript quantities. Either by direct inclusion or as external SC, fragmented 2i SC reactions had efficiencies similar to those observed in cDNA pools and therefore proved suitable for the reliable and accurate absolute quantification of *HBB* mRNA variants in primary samples.

Of note, we observed that the efficiencies for normal *HBB* were slightly but consistently higher (3.32 ± 1.71%) than for the aberrant *HBB* in all the different SC and cDNA serial dilutions, a likely effect of the smaller size for the normal compared to the aberrant *HBB* template.

The current methodology may serve as a template for the quantification of alternative splicing or of countless other mutations that bring about disease through aberrant splicing. Routine assessment of the latter, including the common thalassemic *HBB^IVSII−654(C > T)^* (HBB:c.316-197C > T) mutation [27], by conventional RT-PCR methods has limited utility, whereas assessments of splice events by targeted or global RNA-seq may impose unsuitable technical, cost or material requirements for many applications [28,29]. Adoption of the current assay would be simple and cost-effective and would allow clear conclusions on the regulation of alternative splicing or on the effect of therapeutic interventions, including pharmacological treatments, DNA editing or targeted mRNA knock-down, and on molecular mechanisms underlying their action. An advantage to other PCR-based methods of the current method for relative quantitation of different transcript variants is its independence from measurements of endogenous housekeeping-gene cDNAs or panels [30]. Therefore, this likewise allows the accurate comparison of variant ratios between samples without additional internal reference. Conversely, comparison between samples for absolute quantities of normal and aberrant splice variants would have to take into account the level of erythroid differentiation, the variation of which in culture can be addressed by normalizing the total *HBB* transcript levels to endogenous *HBA* expression, based on the 2 ^−(ΔΔCt)^ method [31]. Similarly, assessment of normal and aberrant *HBB* relative expression can be performed via the 2 ^−(ΔΔCt)^ method between non-treated and treated patient-derived samples or relative to normal samples (wtHBB/HBA and IVSI-110/HBA). In addition to the SC-based absolute quantification method, measurement of the percentage of correctly spliced *HBB* mRNA can be performed by adapting the 2 ^−(ΔΔCt)^ method to use normal *HBB* mRNA as the gene of interest and the total *HBB* mRNA as the reference gene. However, for this application of the 2 ^−(ΔΔCt)^ method (wtHBB/Total HBB) and in contrast to SC-based measurements, normal control samples need to be included to serve as positive controls (100% correct splicing of *HBB* mRNA) for estimation of the percentage of correctly spliced *HBB* mRNA out of the total *HBB* mRNA variants. Of note, all three above applications of the 2 ^−(ΔΔCt)^ method rely on ΔCt calculations between globin genes only, which is critical to accuracy. Importantly, the 2 ^−(ΔΔCt)^ method is sensitive even to slight differences in primer efficiencies, in particular when the difference in Ct values between test and reference genes are large (see [32] and references therein), which multiplies any differences in efficiency exponentially. This usually poses a problem for analyses of the exceptionally highly expressed globins in the erythroid lineage, but not when *HBA* is used as the endogenous calibrator for the analysis of *HBB* expression. While this in turn makes the comparison between normal and patient samples blind to comprehensive effects of the disease-causing mutation on erythroid differentiation, there is by definition no endogene that shows invariable expression across different stages of erythroid development that would not at some stage of erythroid development introduce unacceptably high measurement errors for analyses of *HBB* expression with the 2 ^−(ΔΔCt)^ method.

In vitro differentiation allows analyses of patient-specific gene expression even for transfused patients, but it is an imperfect representation of globin expression in PB [33]. Likewise, this study showed a difference in HBB/HBA protein expression ratios in culture compared to PB, which is of interest in its own right, but also regarding the extrapolation of our in vitro findings to in vivo globin expression. The observed discrepancies may most likely be brought about by differences in promoter activity, which might not affect the relative expression of aberrant and normal transcripts from the same gene. However, splicing, and thus absolute and relative *HBB* transcript levels, might also differ in PB from those in our cell culture model. However, without access to untransfused *HBB^IVSI−110^*-homozygous samples, this hypothesis cannot be tested.

Absolute quantification in principle allows clear-cut comparisons of the number of variant-specific mRNA molecules per cell. We can thus calculate that our measurements indicate expression of 7 ± 8 molecules of aberrant and 11 ± 10 molecules of normal *HBB* mRNA per cell in 3-day cultures for *HBB^IVSI−110^* patients, compared to 298 ± 52 molecules of normal *HBB* mRNA per cell in equivalent cultures from healthy controls. The correspondingly large differential of in vitro *HBB* mRNA expression between patient and normal cells, by a factor of 16 for total *HBB* mRNA and a factor of 27 for normal *HBB* mRNA, is tantalizingly large but reflects contribution of a combination of factors. Of biological interest as underlying factors are differences in the efficiency of RNA processing, as well as marked differences in the in vitro differentiation of normal and thalassemic cells. However, uncertainties concerning lossless collection of initial cell material, quantitative extraction of RNA and quantitative conversion of mRNA to cDNA also enter into this differential as technical factors bound up not only with RT-qPCR methodology but at least in part with any alternative technology of suitable sensitivity.

Finally, our correlation analyses of HBB transcripts and protein levels tied in with the established negative role of aberrant *HBB* transcripts in HBB production [13] and with their causative role in the non-proportional increase of HBB/HBA chain levels with the increase of the functional *HBB* mRNA. Moreover, based on the novel multiplex RT-qPCR protocol, this study revealed that aberrant RNA constitutes 60.9 ± 7.4% of total *HΒΒ*-derived mRNA in primary erythroid cell cultures of *HBB^IVSI−110^* patients, with residual expression of (differentiation-normalized) normal *HBB* mRNA reaching 37.2% × 2.1 ^±1^ of levels observed in corresponding healthy cultures. This first accurate measurement of missplicing and of normal *HBB* mRNA amounts in *HBB^IVSI−110^*-homozygous erythroid cells underpins our previous observations that absolute *HBB* mRNA levels are not limiting for HBB production and that removal of aberrant transcripts alone may be sufficient to achieve a therapeutic effect in *HBB^IVSI−110^* patients [13,14,15], even in the absence of a functional *HBB* trans- or endogene [13].

## 4. Materials and Methods

### 4.1. Subjects

Research programs and protocols for the use of human hematopoietic stem cells were approved by the Cyprus National Bioethics Committee (Applications ΕΕΒΚ/ΕΠ/2012/02 “Advancing Gene Therapy Vectors for Thalassaemia,” ΕΕΒΚ/ΕΠ/2013/23 “ThalaMoSS” and ΕΕΒΚ/ΕΠ/2018/52 “βThal-GT4U”). All study participants gave written informed consent for research use of their blood samples and were tested for additional mutations in the *HBA* and *HBB* genes in line with routine laboratory procedures [34]. Participation in this study did not affect the treatment of study subjects.

### 4.2. Oligonucleotides and Probes

All oligonucleotide primers and ZNA probe were synthesized by Metabion, the TagMan MGB probe by Life Technologies. Sequences of primers and probes are listed in Table 2.

### 4.3. Plasmid Production

All constructs were produced in One Shot TOP10 chemically competent *E. coli* bacteria in Lowry Broth (Sigma-Aldrich) medium with antibiotic selection (ampicillin, 100 μg/mL). Constructs were extracted using the NucleoBond Xtra Maxi kit (Macherey Nagel GmbH, Düren, Germany), according to the manufacturer’s instruction, and DNA was dissolved in HPLC water. Plasmid DNA concentration was measured by UV absorbance using a ND-1000 Spectrophotometer (Thermo Fisher Scientific, Carlsbad, CA, USA).

### 4.4. Plasmid-Based Standard Curve

Relative and absolute quantification of normal and aberrant *HBB* transcripts rely on standard curves based on plasmids harboring cDNA sequences that surround the *HBB^IVSI−110^* mutation site. We tested either a combination of two plasmids that harbor a single cDNA fragment (1-insert (1i) constructs) of the normal (pCR2.1_HBB_N) and aberrant (pCR2.1_HBB_A) *HBB* transcript, respectively, or a single plasmid (pCR2.1_HBB_N+A), containing both template cDNA sequences in tandem (2-insert (2i) constructs). For their construction, cDNA fragments amplified by primer pair IVSI-110_FW/IVSI-110_RV to produce a 309-bp normal and a 328-bp aberrant product were initially cloned into the two separate 1i vectors, pCR2.1_HBB_N and pCR2.1_HBB_A, respectively, using the TA Cloning Kit (Invitrogen^™^, Thermo Fisher Scientific, Carlsbad, CA, USA) according to the manufacturer’s instructions. Subsequently, the pCR2.1_HBB_A-encoded cDNA was PCR-amplified using primers tagged with EcoRV and XhoI restriction sites in order to allow directional cloning of the amplicon into pCR2.1_HBB_N, yielding the final 2i pCR2.1_HBB_N+A plasmid. All three constructs were confirmed by sequencing across all ligation junctions.

Overall, five different standard curves (SCs) were produced of 6-log serial construct dilutions, starting from 2.0075 × 10^6^ molecules down to two molecules/μL, corresponding to 10^−1^ to 10^−6^ ng/μL. The first two SC were produced by using the pCR2.1_HBB_N and pCR2.1_HBB_A 1-insert plasmids separately (circular 1i simplex SC) or together (circular 1i duplex SC) in a 1:1 ratio. The rest of the SCs were produced by using the 4545-bp 2-insert plasmid pCR2.1_HBB_N+A in different structural conformations, (i) circular (Circular 2i SC), (ii) linear after SpeI restriction digest (Linear 2i SC) and (iii) fragmented after EcoRI/XbaI sequential restriction digest (Fragmented 2i SC). Briefly, 2 μg of pCR2.1_HBB_N+A construct was used in each restriction reaction using the appropriate NEB buffer for each restriction enzyme. Plasmids were incubated at 37 °C for 16 h for each digestion and complete restriction digest followed by heat inactivation at 65 °C for 20 min and conformation of the constructs confirmed by agarose gel electrophoresis.

All 10-fold serial dilution SCs (Circular simplex and duplex 1-insert SCs, circular, linear and fragmented 2-insert SCs) were prepared in independent triplicates and of each dilution, and 2 μL was used per reaction. To facilitate third-party research of *HBB^IVSI−110^* thalassemia, we have made plasmids pCR2.1_HBB_N, pCR2.1_HBB_A and pCR2.1_HBB_N+A available via the Addgene plasmid repository (#87,847, # 87,848 and # 87,849; see also Appendix A “Plasmids”, holding Genbank sequences and plasmid maps for all three plasmids). Based on pCR2.1_HBB_N+A, an additional two 2i SCs were produced by restriction digest (2.5 unit per 1 μg plasmid in 100 μL, following the manufacturer’s protocol), as described in the Results, and employed without further processing in qPCR.

### 4.5. Cell Culture

Samples of between 7 and 25 mL of naïve peripheral blood were collected during pre-transfusion analysis in ethylenediaminetetraacetate-anticoagulated vacuum tubes and stored at room temperature until processing for the isolation of mononuclear cells by phase separation (Lymphocytes, Axis-Shield Diagnostics Ltd., Oslo, Norway) and enrichment of CD34^+^ hematopoietic stem and progenitor cells (HSPCs) by magnetic-assisted cell selection (Miltenyi Biotec, Bergisch Gladbach, Germany) according to the manufacturer’s instructions. Cell expansion, cryopreservation (in 50% fetal bovine serum, 40% culture medium and 10% dimethyl sulfoxide) and erythroid differentiation conditions followed procedures described elsewhere [18].

### 4.6. Quantification of mRNA Expression by RT-qPCR

#### 4.6.1. Reverse Transcription PCR (1st Step)

RNA was extracted from 5 × 10^6^ in vitro differentiated primary erythroid cells using TriZol^TM^ (Invitrogen^TM^, Thermo Fisher Scientific, Carlsbad, CA, USA) and quantified spectrophotometrically. One μg of the total RNA was treated with 0.5 units of DNase I (Invitrogen^™^, Thermo Fisher Scientific, Carlsbad, CA, USA) according to the manufacturer’s instructions. Complementary DNA was synthesized using the TaqMan Reverse Transcription PCR kit (Applied Biosystems, Thermo Fisher Scientific, Carlsbad, CA, USA) following the manufacturer’s instructions based on a 250-ng aliquot of DNase-I-treated RNA in 10 μL reaction volume. The resulting cDNA samples were diluted three-fold with RNase-free water (Sigma-Aldrich, Munich, Germany) to a final cDNA concentration equivalent to 8.33 ng/μL total RNA, and 2 μL were used for each qPCR reaction.

#### 4.6.2. Quantitative PCR (2nd Step A)

Gene expression quantification was performed using the 7900HT Fast Real-Time PCR System (Applied Biosystems) and probe-based duplex qPCR using the Multiplex PCR Kit (Qiagen) according to the manufacturer’s instructions and using the following qPCR cycling conditions (50 °C for 2 min, 95 °C for 15 min, 40 cycles of 95 °C for 30 s and 60 °C for 1 min). Detection was based on primer pair HBB_Ex1_FW/HBB_Ex2_RV (900 nM each)) and probes A_MGB_VIC for aberrant and N_ZNA_FAM for normal cDNA) (250 nM each). Two μL of standard curve and sample cDNA serial dilutions or 8.33 ng/μL sample cDNA were used per 12.5 μL reaction. Each sample was run in triplicate, and each run included three independently prepared SCs of plasmid pCR2.1_HBB_N+A, as described above, and a non-template control (NTC) in triplicate. The plasmid SC allowed absolute quantification of each splice variant. Percentage of correct and aberrant splicing was measured as follows:

##### Transcript Percentage Contributions


*Percentage of Correct Splicing = 100 × (abs quant. of Normal HBB/abs quant. of (Normal HBB + Aberrant HBB))*



*Percentage of Aberrant Splicing = 100 − (% of correct splicing)*


#### 4.6.3. Quantitative PCR (2nd Step B)

Relative expression levels of probe-based values for normal and aberrant *HBB* mRNA and for SYBR-Green-based values for total *HBB* mRNA was assessed via the 2 ^−(ΔΔCt)^ method, where *HBA* expression was used as erythroid differentiation reference relative to the average ΔΔCt value of all untreated patient samples or of healthy controls. Measurement of endogenous *HBA* levels allowed normalization of *HBB* expression for the level of erythroid differentiation based on the 2 ^−(ΔΔCt)^ method [31].

##### Relative Expression of Normal, Aberrant and Total *HBB* Variants (2 ^−(ΔΔCt)^ Method)


*ΔCt (Sample) = Ct HBB variant − Ct HBA*


*Expression ratio test* vs. *control = 2 ^−(ΔCt (Test Sample − ΔCt (Control Sample *))^*


** Untreated patient control or healthy control*


In addition to the SC-based absolute quantification, the percentage of correct splicing was assessed by using the relative quantities of the Normal *HBB* (probe-based qPCR described above) to the total *HBB* (SYBR Green), in patient samples compared to the healthy controls (2 ^−(ΔΔCt)^ method).

##### Percentage of Correct Splicing (2 ^−(ΔΔCt)^ Method)


*ΔCt (Sample) = Ct _Normal HBB_ − Ct _Total HBB_*



*Percentage of correct splicing = 100 × (2 ^−(ΔCt (Patient Sample) − ΔCt (Healthy Sample^)*


Briefly, quantification was performed using the SYBR Green PCR Master Mix (Applied Biosystems) according to manufacturer’s instructions and using standard qPCR cycling conditions (50 °C for 2 min, 95 °C for 10 min, 40 cycles of 95 °C for 15 s and 60 °C for 1 min). The annealing/extension step for *HBB* and *HBA* reactions was set to 65 °C for 1 min to avoid non-specific detection of the murine *Hbb* transcripts or the formation of primer-dimer artefacts, respectively. All SYBR Green reactions were performed using each set of primers (300 nM) and 8.33 ng/μL sample cDNA per 25-μL reaction.

PCR efficiencies of each primer pair was assessed as follows. For human genes expression, *HBB* variants (Probes), total *HBB* and *HBA* (SYBR green), an SC of 10-fold serial dilutions was prepared, starting from a mixture of cDNA (equivalent of 16.66 ng RNA) of cultures of *HBB^IVSI−110^-homozygous* CD34^+^ cells on day 3 of induced differentiation.

### 4.7. Alternative Transcript Quantifications

Semi-quantitative end-point PCR (HotStart Q5, New England Biolabs Inc., Ipswich, MA, USA) employed 25 ng cDNA in 50-μL reactions) and 10 ng control plasmids (pCR2.1_HBB_N, pCR2.1_HBB_A) for 25, 30 and 35 cycles, before further analyses. Quantification of normal and aberrant transcript amounts by integration of fluorescent band intensities (SafeRed, GeneCopoeia, Rockville, MD, USA) was performed using ImageJ [36] after agarose gel separation and purification (QIAquick PCR Purification kit, Qiagen, Hilden, Germany). In parallel, 50 ng of PCR products were analyzed by cycle sequencing (BigDye v1.1, Applied Biosystems, Thermo Fisher Scientific, Carlsbad, CA, USA) and decomposition of sequence traces using the TIDER [19] HDR quantification algorithm and the ICE knock-in score algorithms [20], providing as wild-type control the sequence trace for plasmid pCR2.1_HBB_N, and as sham input for the initial detection of the aberrant sequence insert a suitable gRNA sequence (GTGGTGAGGCCCTGGGCAGG) and HDR donor sequence (GGTGGTGAGGCCCTGGGCAGTCTATTTTCCCACCCTTAGGCTGCTGGTGGTCTACCCTT).

### 4.8. High-Performance Liquid Chromatography

Reversed-phase high-performance liquid chromatography (HPLC) analyses were performed using a 25-cm Jupiter C18 column (Phenomenex Inc., Torrance, USA) and a linear low-pressure gradient of either 2:1 (*v*:*v*) acetonitrile:methanol or pure acetonitrile against 0.1% TFA on a Prominence system (Shimadzu Corporation, Kyoto, Japan) with an SPD-M20A diode array detector. The equivalent of 0.8 μL peripheral blood or of 500,000 primary erythroid cells collected after seven days of in vitro differentiation were analyzed per sample run. HBB expression is expressed as a ratio of HBB peak area to same-sample HBA peak area or as a percentage of the HBB/HBA ratio in normal controls. Similarly, the proportion of the different HBB-like globin chains (HΒΒ, HBD, HBG1 and HBG2) in each sample is given as the percentage of each HBB-like globin peak area to the total area of all HBB-like globin peaks.

### 4.9. Bioinformatics and Statistical Analyses

Primers and probes were designed based on NCBI transcript sequence NM_000518.4 (*HBB*), mRNA sequence information from Vadolas et al. [4] (*HBB^IVSI−110^)* and NCBI genomic DNA Reference Sequence NG_000007.3 (*HBB* locus), and were tested for self-annealing and hairpin structures using Vector NTI 11 software (ThermoFisher Scientific, Carlsbad, CA, USA) with default settings. Primer specificity for human sequences was assessed using the primer-BLAST web tool. Data transformation, collection and visualization of summary data was performed using Excel (Office 2010, Microsoft Corporation, Redmond, WA, USA.). Amplification factor and efficiencies were calculated based on the slope of the plasmid- or cDNA-SC (Y axis: ΔRn X axis: Ct) using the online tool qPCR Efficiency Calculator provided by ThermoFisher Scientific [37]. For statistical analyses and visualization of the different SCs in linear regression graphical format, data were transferred to Prism 8.0 (GraphPad Software Inc., La Jolla, CA, USA), from which all SC equations, correlation analyses and slopes were exported, the latter for calculation of efficiencies in the qPCR Efficiency Calculator [37]. Kolmogorov-Smirnov test was used to test for normality, and group-wise comparisons were conducted by parametric or non-parametric tests accordingly. Pearson’s *r* correlation coefficient was computed assuming data were sampled from Gaussian distributions and using a two-tailed *p* value with a 95% confidence interval. Values are shown as arithmetic mean ± standard deviation of the population mean, except for the analysis of relative expression levels using the 2 ^−(ΔΔCt)^ method, where values are shown as the geometric mean multiplied/divided by the geometric standard deviation (×σ_g_
^±1^) [38].

## Figures and Tables

**Figure 1 ijms-21-06671-f001:**
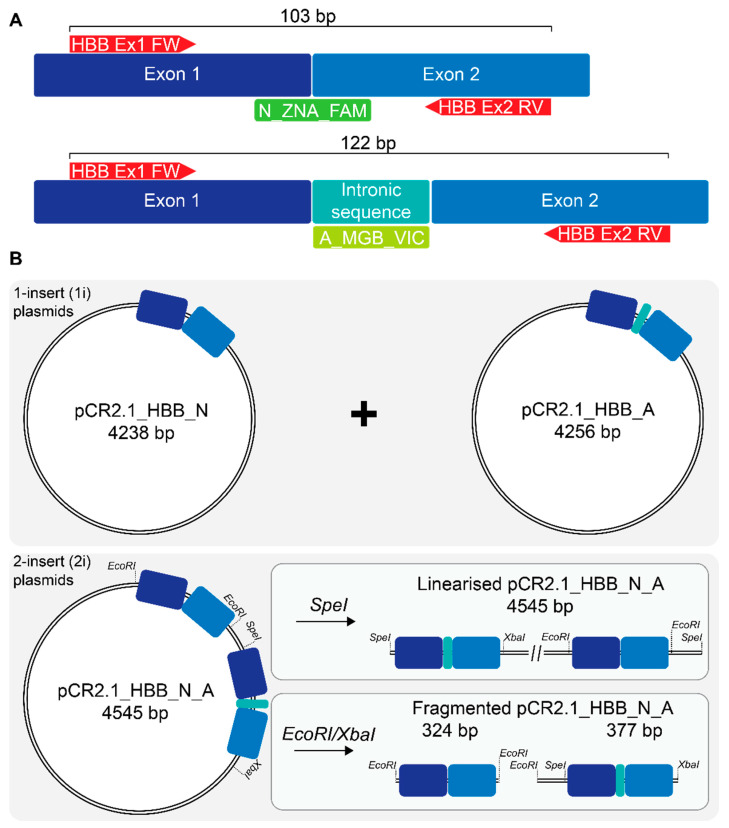
Primers, probes and standard-curve plasmids for the duplex RT-qPCR assay. (**A**) The relevant section of normal or aberrantly spliced β-globin mRNA is depicted and labeled for its key components. IVSI-110_FW and _RV are primers common to the normal and the aberrant amplicon, while probes N_ZNA_FAM and A_MGB_VIC, respectively, specifically detect and label normal and aberrant β-globin mRNA. (**B**) Either a combination of 1-insert (1i) plasmids pCR2.1_HBB_N or pCR2.1_HBB_A in defined molecular ratios or 2-insert (2i) plasmid pCR2.1_HBB_N+A, which holds tandem copies of normal and aberrant cDNA fragments, were used as the basis for standard curve and RT-qPCR-based quantification of both transcripts. Purple rectangle—exon 1 in; blue rectangle—exon 2; light blue rectangle—aberrantly retained intronic sequence (19 nt). Components are not depicted to scale.

**Figure 2 ijms-21-06671-f002:**
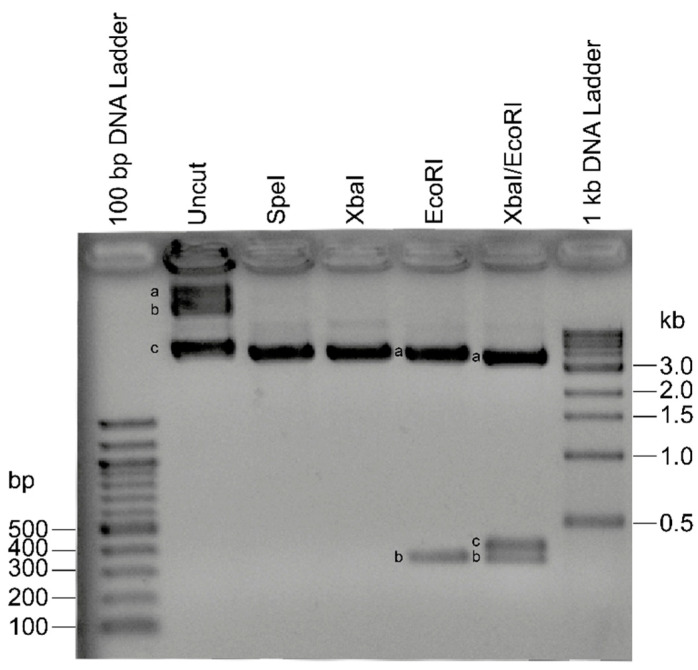
Comparison of the different forms of the pCR2.1_HBB_N_A 2-insert construct with and without restriction digest. Linearized and fragmented 2-insert standard curves are based on serial dilution after restriction digest with SpeI and double digest with EcoRI/XbaI, respectively. Uncut: (**a**) multimeric supercoiled circular constructs, (**b**) nicked, relaxed circular constructs, (**c**) supercoiled construct; SpeI: linearized construct (4545 bp); XbaI: linearized construct (4545 bp); EcoRI: (**a**) pCR2.1 backbone plus IVSI-110 fragment (4221 bp), (**b**) wtHBB fragment; EcoRI/XbaI: (**a**) pCR2.1 backbone (3844 bp), (**b**) IVSI-110 fragment (377 bp); (**c**) wtHBB fragment (324 bp).

**Figure 3 ijms-21-06671-f003:**
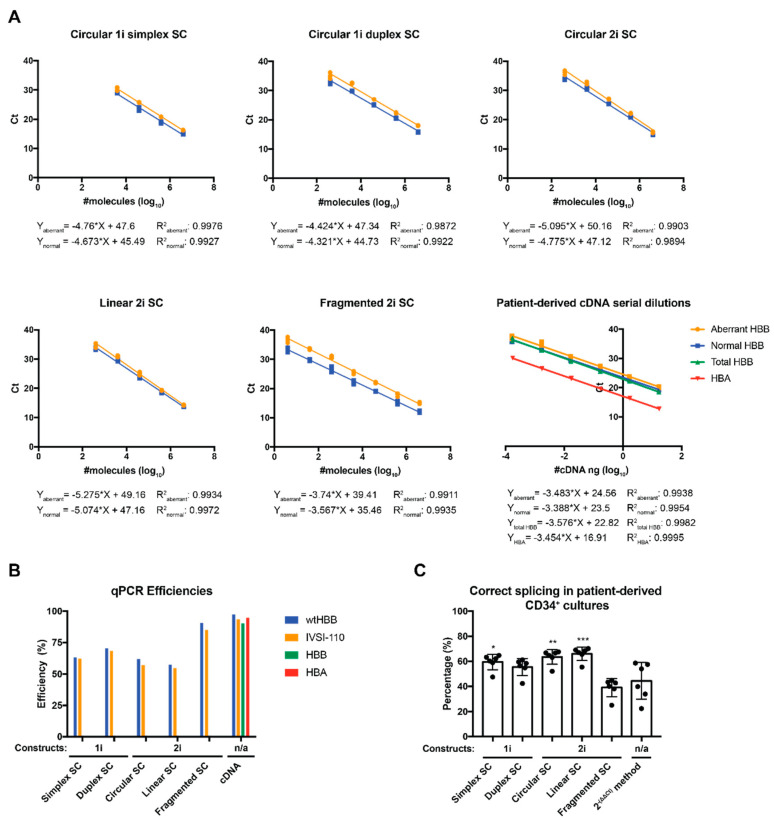
Evaluation of assay performance with standard curves and samples (**A**) Graphical Illustration of the different standard curves (circular 1-insert (1i) simplex and duplex standard curves (SCs) pCR2.1_HBB_N and pCR2.1_HBB_A constructs; circular, linear and fragmented (2i) pCR2.1_HBB_N_A 2-insert construct) used for the absolute quantification of normal and aberrant *HBB* mRNA variants (Multiplex probe-based RT-qPCR) and of patient-derived cDNA serial dilutions (10-fold) for the assessment of quantitative PCR reaction efficiencies for normal and aberrant *HBB* mRNA variants (Multiplex RT-qPCR), total *HBB* and *HBA* (SybrGreen). Each graph includes the predicted equations and R-squared (R^2^) of each SC or cDNA serial dilution. (**B**) Comparison of calculated efficiencies of the multiplex probe-based reactions using different SCs with those obtain with cDNA serial dilutions. (**C**) SC equations were used for the absolute quantification of percentage of correctly and aberrantly spliced *HBB* mRNA. Percentage of correct splicing was assessed with the 2 ^−(ΔΔCt)^ method, where Ct values for the normal *HBB* and total *HBB* were used as gene of interest and reference, respectively, and as reference control the average value of Normal (*n* = 2) samples. One-way ANOVA was used for the multiple comparison of the percentages of correct splicing on day 3 of induced erythroid differentiated patient-derived CD34+ cells (*n* = 6) between those calculated using the different SC equations with those measured via the 2 ^−(ΔΔCt)^ method. * *p* value 0.0114; ** *p* value 0.0017; *** *p* value 0.0005.

**Figure 4 ijms-21-06671-f004:**
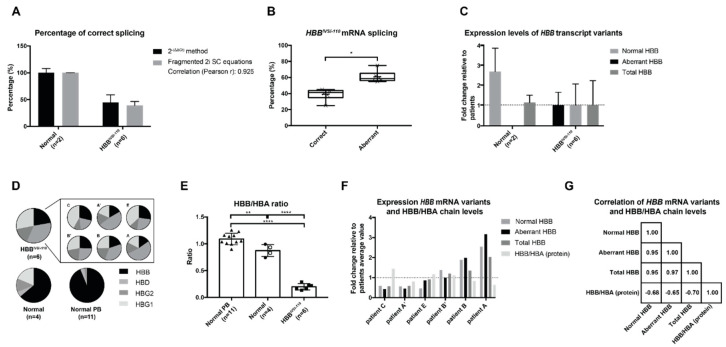
Analysis of in vitro differentiated normal and patient-derived CD34^+^ cells at the RNA and protein level. (**A**) Direct comparison of percentages of correct splicing in normal cells (*n* = 2; SC: 100.0 ± 8.1% vs. 2 ^−(ΔΔCt)^ method: 100.0 ± 1.1%) and patient-derived cells (*n* = 6; SC: 44.4 ± 14.7% vs. 2^-(ΔΔCt)^ method: 39.1 ± 7.3%) calculated by using the 2 ^−(ΔΔCt)^ method and absolute quantification method using the fragmented 2i SC equations. Correlation of percentage of correct splicing with the 2 ^−(ΔΔCt)^ method vs. fragmented SC equations, Pearson r: 0.925; *p* value 0.001. (**B**) Percentage of splicing on day 3 of induced differentiated patient-derived cultures (*n* = 6), as calculated based on the 2i SC method. Aberrantly spliced transcripts are significantly more abundant than normal transcripts (* *p* value 0.0146 by paired t test). (**C**) Relative expression levels of the normal, aberrant and total *HBB* mRNA variants in normal and patient-derived cultures, normalized to *HBA* mRNA levels (reference gene) and using as reference patient-derived average values. (**D**) Proportion of HBB-like globin chains in normal PB (*n* = 11) and on day 7 of induced differentiated normal (*n* = 4) and patient-derived (*n* = 6) cultures, including individual distributions for patient cultures. Patients A and B were each analyzed by two independent cultures (A, A’ and B, B’, respectively). (**E**) HBB/HBA ratios for samples detailed in panel D: ** *p* value 0.005; **** *p* value < 0.0001 by ordinary one-way ANOVA with Tukey correction for multiple comparisons of column means. (**F**) Illustration of the relative expression levels of the normal, aberrant and total *HBB* mRNA variants normalized to *HBA* mRNA levels and of HBB/HBA globin chain ratios in six patient-derived cell cultures (see labels in panel D) as a fold change relative to patient-derived average values. (**G**) Correlation (Pearson r) analysis of the fold change expression levels (2 ^−(ΔΔCt)^ method) of normal, aberrant and total *HBB* mRNA levels, normalized to *HBA* mRNA and HBB/HBA globin chain ratios in patient-derived cultures (*n* = 6), relative to patient-derived average values.

**Table 1 ijms-21-06671-t001:** Characteristics of the probe-based multiplex reactions using standard curves of circular pCR2.1_HBB_N and/or pCR2.1_HBB_A 1-insert (1i) constructs separated (simplex) or together in 1:1 molar ratio (duplex) or of different conformations (circular, linear and fragmented) of the pCR2.1_HBB_N_A 2-insert (2i) construct, and of SYBR Green reactions using serial dilutions of cDNA derived from a cell culture pool (*n* = 6) of induced differentiated (day-3) patient-derived CD34^+^ cells.

Standard Curve (SC) Composition	SC Equation	R^2^	Efficiency
pCR2.1_HBB_N (Circular 1i simplex SC)	Y = −4.673 × X + 45.49	0.9927	63.28
pCR2.1_HBB_A (Circular 1i simplex SC)	Y = −4.76 × X + 47.61	0.9976	62.21
pCR2.1_HBB_N (Circular 1i duplex SC)	Y = −4.321 × X + 44.73	0.9922	70.38
pCR2.1_HBB_A (Circular 1i duplex SC)	Y = −4.424 × X + 47.34	0.9872	68.28
pCR2.1_HBB_N (Circular2i SC)	Y = −4.775 × X + 47.12	0.9894	61.97
pCR2.1_HBB_A (Circular 2i SC)	Y = −5.095 × X + 50.16	0.9903	57.13
pCR2.1_HBB_N (Linear 2i SC)	Y = −5.074 × X + 47.16	0.9972	57.43
pCR2.1_HBB_A (Linear 2i SC)	Y = −5.275 × X + 49.16	0.9934	54.73
pCR2.1_HBB_N (Fragmented 2i SC)	Y = −3.567 × X + 35.46	0.9935	90.70
pCR2.1_HBB_A (Fragmented 2i SC)	Y = −3.74 × X + 39.41	0.9911	85.09
Normal *HBB* cDNA sample dilution	Y = −3.388 × X + 23.5	0.9954	97.31
Abnormal *HBB* cDNA sample dilution	Y = −3.483 × X + 24.56	0.9938	93.69
Total *HBB* cDNA sample dilution	Y = −3.576 × X + 22.82	0.9982	90.39
*HBA* cDNA sample dilution	Y = −3.454 × X + 16.91	0.9995	94.77

Y—threshold cycle, X—log of target quantity.

**Table 2 ijms-21-06671-t002:** Probe and primer sequences used in the manuscript.

Primer	Sequence (5′–3′)
IVSI-110_FW	TTCACTAGCAACCTCAAACAGACACC
IVSI-110_RV	CACAGTGCAGCTCACTCAG
HBB_Ex1_FW	GGGCAAGGTGAACGTG
HBB_Ex2_RV	GGACAGATCCCCAAAGGAC
A_MGB_VIC	VIC-TAAGGGTGGGAAAATAGA-MGB
N_ZNA_FAM	6-FAM-TGG G(PDC)A GG(PDC) TG(PDC) TG-ZNA-3-BHQ-1
HBA FW	GGTCAACTTCAAGCTCCTAAGC [35]
HBA RV	GCTCACAGAAGCCAGGAACTTG [35]

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
