# Peer review of "Relative and Absolute Quantification of Aberrant and Normal Splice Variants in *HBB^IVSI−110 (G > A)^* β-Thalassemia"

_ijms, 2020, doi:10.3390/ijms21186671_

Round 1

Reviewer 1 Report

I have no major concerns regarding the manuscript.

Author Response

We thank Reviewer 1 for the positive evaluation and the swift assessment of our manuscript.

Reviewer 2 Report

The subject of this manuscript is one which some may have thought went out of fashion a couple of decades ago, however it still is relevant for current methods of gene manipulation.

The authors have done a very much updated analysis of the exact amount of mRNA and Hb production is produced from a mutant allele which is common in the Mediterranean region, IVS1-nt 110. This is a severe Beta plus mutation is well known to those of us in the field, since many patients carry this as compound heteroxygotes or homozygotes. The patients have transfusion dependent disease and the mechanism of the mutation is known to be the formation of a new splice site which produces an abnormal mRNA and very reduced amounts of Hb. These findings date back to the 1980's and the current relevance is to prove how much the mutant RNA disrupts the normal Hb synthesis so as to see if knocking down the mutant allele (for instance with a Crisper system) has potential for gene therapy for patients carrying a particular mutation.

The authors first perform a RT PCR system with an internal quantitative control using a plasmid construct with the mutant and wild type alleles on the same plasmid. They perform the PCR on supercoiled plasmid and linearized and also with the specific mutant/wt template DNA excised by restriction enzymes. The fragment has better PCR efficiency. They adjust the PCR with minute precision (reagent type, cycle number etc) so as to be able to see the number of copies of mRNA of each allele, mutant and wild type, comparing the efficiency of the PCR for each template. This is sort of, well....tedious but the end result is a system with a standard curve that can quantify in an accurate comparative fashion the amounts of mutant and wild type RNA. The authors then perform erythroid cultures from peripheral blood progenitors and measure the IVS 110 mutant and wild type RNA and hemoglobins produced. The mutant allele produces less mRNA however it produces MUCH less protein product, which the authors deduce means the abnormal RNA acts like a dominant negative and therefore inactivating the mutant allele in these patients would allow (theoretically) these patients to produce normal Hb. 

In general this study is of course not completely original, as nearly 3 decades ago, scientists were trying to determine the nature of the detrimental action of malignant alleles. However the precision with which it is done, and the actual proof of how low the normal protein product (Hb) is relative to the amount of RNA is new. Indirect proof was provided years ago by Hb electrophoresis in nontransfused patients in which some HbA could be detected but very very little, and most patients are heavily transfused so the authors used progenitor cultures derived from peripheral blood and not patient retic RNA samples, for instance.

All in all this study is nicely done. I have a few questions for clarification and a few suggestions to improve the data presentation a bit.

  1. Top of page 7. Progenitor cultures are used for the reason stated above. The authors should verify that irradiation of the transfused unit really DOES kill the erythroid progenitors in the unit otherwise it is possible that some of the normal splicing and normal Hb product are from the transfused unit. This reviewer knows that the dose that is used to irradiate transfused blood is the amount needed to kill lymphocytes to avoid transfusion associated GVHD. I do not know how much it takes to kill peripheral blood RBC progenitors.
  2. In truth I think that the authors could have done the study using reticulocyte RNA. This disappears within 2 days of the blood being donated and by the time the blood unit is given (after hepatitis  testing), the retic RNA is no more so donor retic RNA would not be a problem. However the amount of retic RNA is smaller than what was gotten from erythroid cultures so that approach makes sense.
  3. The authors state that cultures were derived from 4 patients, and there were 6 cultures. Figure 4F shows patients 1-6 this should be labelled patients 1-4 and those with multiple cultures should be noted "patient 1, culture 1 and culture 2). This is important since there is some variability among the cultures and it is interesting to see if the cultures were reproducible between themselves. For instance culture 6 had a lot of RNA. Was this duplicated in another culture?
  4. One more point about this figure (it is an important part of the results). FIgure 4 D shows a pie graph including all 6 cultures but I think there is a better way to graphically present the graphs. I would say to eliminate the pie graph and instead use "whisker plots" for each type of Hb that the cells produce: HbA, HbA2, HbF and aberrant Hb. There should be one for each culture separately and not the way it is that all 6 cultures are combined in one pie graph.
  5. In summary this is a nice study which took nearly 30 years to perform since the original methodology was done. The methodology is appropriate to any other type of splicing mutation and therefore is widely applicable. The findings can open a window to try gene therapy for inactivating the "domnant negative" transcripts.

Author Response

[Please, see our responses below in yellow highlights.]

> We thank Reviewer 2 for the positive evaluation and the substantial time taken to assess our manuscript and summarise its finding and relevance.

1. Top of page 7. Progenitor cultures are used for the reason stated above. The authors should verify that irradiation of the transfused unit really DOES kill the erythroid progenitors in the unit otherwise it is possible that some of the normal splicing and normal Hb product are from the transfused unit. This reviewer knows that the dose that is used to irradiate transfused blood is the amount needed to kill lymphocytes to avoid transfusion associated GVHD. I do not know how much it takes to kill peripheral blood RBC progenitors.

> This is a highly perceptive comment, even more so because transfused blood is not routinely irradiated in Cyprus, and relies on leucapheresis, only, albeit with a stringent quality threshold of <1x10^6 WBC/unit (measured as CD45+ cells). This low carryover of WBCs and our utilisation of recipient blood between 10 to 15 days after the last transfusion cannot exclude but reduces to insignificant background any potential contribution of donor HSPCs to the initial sample. Persistence of nucleated donor cells in transfused patients is short (up to six days for packed RBC with 109 WBC per unit, PMID 1627807 Table 1) and for transfusions with comparable quality thresholds (<1x10^6 WBC/per unit) results in qPCR-detectable nucleated cells in only a small minority of transfused patients even during the first 10 days after transfusion (PMID 31518003 Figure 2). We have added the information that we used samples “collected during pre-transfusion analysis” to section 4.5 (line 451) of the manuscript and if requested would be happy to add any of the above arguments to the Discussion or to section lines 196-202 of our results.

2. In truth I think that the authors could have done the study using reticulocyte RNA. This disappears within 2 days of the blood being donated and by the time the blood unit is given (after hepatitis  testing), the retic RNA is no more so donor retic RNA would not be a problem. However the amount of retic RNA is smaller than what was gotten from erythroid cultures so that approach makes sense.

> Our wider research focus is the gene-therapy treatment and analysis of haematopoietic stem cells, so that the choice of CD34+-derived cell cultures for RNA quantification tied in with our reliably established methodology and with our own envisaged use of the established protocol. However, the reviewer’s suggestion of employing reticulocyte RNA in our analyses of HBBIVSI-110(G>A) in order to get insights into in vivo expression patterns while avoiding detection of donor-derived expression prompted us to study protocols for selective reticulocyte isolation and RNA half-life. The methodology is currently not established in our laboratory and will need alternative gradient reagents and procedures to those currently employed. However, the approach might indeed give adequate RNA quantities and quality for in vivo insights and while out of scope for the current manuscript will be tested for our subsequent investigations. We are grateful for this suggestion. 

3. The authors state that cultures were derived from 4 patients, and there were 6 cultures. Figure 4F shows patients 1-6 this should be labelled patients 1-4 and those with multiple cultures should be noted "patient 1, culture 1 and culture 2). This is important since there is some variability among the cultures and it is interesting to see if the cultures were reproducible between themselves. For instance culture 6 had a lot of RNA. Was this duplicated in another culture?

> We thank the reviewer for this correction of what would have been incomplete reporting and, moreover, confusing to the reader. We have now indicated same-patient cultures in Figure 4F and remarked this in the legend text accordingly: “Patients A and B were each analyzed by two independent cultures (A, A’ and B, B’, respectively).” RNA and protein parameters are highly variable for independent same-patient cultures, which does not allow any further conclusions from Figure 4F.

4. One more point about this figure (it is an important part of the results). FIgure 4 D shows a pie graph including all 6 cultures but I think there is a better way to graphically present the graphs. I would say to eliminate the pie graph and instead use "whisker plots" for each type of Hb that the cells produce: HbA, HbA2, HbF and aberrant Hb. There should be one for each culture separately and not the way it is that all 6 cultures are combined in one pie graph.

> We agree with the sentiment of displaying all cultured thalassemic cells individually. Box-and-whisker plots for each culture would not work (there would be only one dot per category), and one box-and-whisker plot (displaying all data points) for all six thalassemic cultures with different symbol colour or shape for each patient turned out to be confusing to the beholder. We have therefore instead opted for individual pie charts for each thalassaemic culture, in addition to the summary pie chart and have resized the panel accordingly. We hope that this is in line with the Reviewer’s suggestion.

5. In summary this is a nice study which took nearly 30 years to perform since the original methodology was done. The methodology is appropriate to any other type of splicing mutation and therefore is widely applicable. The findings can open a window to try gene therapy for inactivating the "domnant negative" transcripts.

> Thank you once more for this assessment and the insightful comments.

Reviewer 3 Report

The authors quantify the transcripts, determine the mRNA ratios and quantities in blood and cell cultures by duplex reverse-transcription quantitative PCR assay. The authors conclude that the method reported here is sensitive and accurate. In general, the manuscript is well written, and the data is interesting. In order to strengthen the claims made, the authors should address the following:

At least one orthogonal validation, such as the traditional PCR-based methods or Northern Blotting, is suggested to be employed in parallel. The data obtained by different methods could be compared.

Table 1. Under the presented condition, the PCR efficiency appears to be low. Did different reaction conditions have been tried to optimize the quantification? In addition, how do the authors explain the different qPCR efficiencies for the normal and aberrant HBB (normal>aberrant)? Did the authors design different primer and probe sets to target the normal and aberrant transcripts to ensure equal and improved efficiency?

Figure 4. The authors investigated the correlation of the HBB transcripts and the protein levels that assessed by Reversed-phase HPLC, which is well appreciated for its function for versatile and inexpensive globin quantification. If an orthogonal method was utilized parallelly to determine the mRNA levels, then the correlation of each mRNA data to protein levels could be analyzed to support the claims made.

Figure 4. please specify which tests were used for each figure panel.

Author Response

[Please, see our responses below in yellow highlights.]

> We thank Reviewer 3 for the positive evaluation and the substantial time taken to assess our manuscript and suggest improvements.

The authors quantify the transcripts, determine the mRNA ratios and quantities in blood and cell cultures by duplex reverse-transcription quantitative PCR assay. The authors conclude that the method reported here is sensitive and accurate. In general, the manuscript is well written, and the data is interesting. In order to strengthen the claims made, the authors should address the following:

At least one orthogonal validation, such as the traditional PCR-based methods or Northern Blotting, is suggested to be employed in parallel. The data obtained by different methods could be compared.

> Prompted by this Reviewer and as the time frame for resubmission allowed, we have employed the originally used cDNAs for a PCR-based analysis of relative transcript quantities, using both (i) limiting cycles for a semi-quantitative analysis based on gel staining of bands and (ii) TIDER/ICE-based analysis of mixed sequence traces with a modification of analyses routinely used for the quantification of genome editing events. While quantifications based on the same end-point PCR correlated well with one another, they did not do so with RT-qPCR-based measurements and generally indicate lower proporitions of aberrant RNA. One reason for this might be the end-point-based quantification after conventional PCR, which in contrast to standard-curve based RT-qPCR does not allow for correction of the size-related lower amplification efficiency for the aberrant transcript. We do not elaborate this point in the main manuscript, where we remark upon the lower detection of aberrant transcripts by end-point-based PCR, and have included a new supplementary figure 3 to present the data.

Incidentally and with apologies, sample quantities required for Northern blotting would have been prohibitive for the current experimental system. Even given sufficient material, the time line for establishing the method for HBBIVSI-110(G>A) sequences as targets may have caused substantial delays to this resubmission.

Table 1. Under the presented condition, the PCR efficiency appears to be low. Did different reaction conditions have been tried to optimize the quantification? In addition, how do the authors explain the different qPCR efficiencies for the normal and aberrant HBB (normal>aberrant)? Did the authors design different primer and probe sets to target the normal and aberrant transcripts to ensure equal and improved efficiency?

> For fragmented (2i) SC and cDNA as template (the last six lines of the Table 1, see also Figure 3B), PCR efficiencies were above 85% and 90%, respectively. In particular the observed efficiencies of 90% to 97% for cDNA templates are in the ideal range (e.g. https://www.gene-quantification.de/national-measurement-system-qpcr-guide.pdf), whereas efficiencies above 80% are in the acceptable range (e.g. https://doi.org/10.1371/journal.pone.0132666). With efficiencies for the test samples (cDNA) in the ideal range, one goal of the current study was to achieve similar efficiencies for corresponding standard curves. As argued in Results lines 121-159 and Discussion lines 278-320, switching from circular to linear and then fragmented templates led to significant improvements of efficiency for the standard curve. Of note, this manuscript spares the reader the preliminary evaluation of different primer pairs to achieve short amplicons that (i) cover the aberrant HBBIVSI-110(G>A) transcript sequence, (ii) avoid detection of the highly sequence similar δ-globin gene and (iii) work well with both probes employed here (other probes with different chemistry had been employed in the past) to differentiate the normal and aberrant exon1-exon2 borders of the β-globin transcript. We could make this point about preliminary analyses in the manuscript, similar to our pointing out initial evaluation of different RT-PCR kits, but fear that it might be a distraction and add excessive detail. Of note, we point out the inherent difference in amplification of normal and aberrant amplicons owing to their size difference in lines 330 to 332. Duplexing specific detection of both fragments with similar efficiencies made a common primer pair with inherently different amplicon sizes an unavoidable design choice.

Figure 4. The authors investigated the correlation of the HBB transcripts and the protein levels that assessed by Reversed-phase HPLC, which is well appreciated for its function for versatile and inexpensive globin quantification. If an orthogonal method was utilized parallelly to determine the mRNA levels, then the correlation of each mRNA data to protein levels could be analyzed to support the claims made.

> The alternative mRNA quantification methods now introduced in supplementary figure 3 are vastly inferior to the RT-qPCR method presented in this paper. We have therefore refrained from calculating additional correlations with HPLC results for those alternative transcript quantities and hope that the Reviewer agrees with this decision. 

Figure 4. please specify which tests were used for each figure panel.

> The revised manuscript now specifies the statistical test performed for each report of p values in Figure 4, after addition of this information for panels B and E.

Reviewer 4 Report

The authors describe a quantitative RT PCR approach to analysis of HBB variant transcipt ratios in thalassemic blood and in vitro differentiated erythroid cells.

The demonstration that the incorporation of the two targets on the same plasmid provides a more reliable standard curve is unsurprising, as is the increased accessibilty of linearised and fragmented plasmid. This is certanly not the first time that these strategies have been used to optimize qRT-PCR approaches. In general, the work is presented in an unusual level of detail more appropriate to a dissertation than to a concise scientific publication. However, this thorough and well-explained account of the technical aspects may indeed be of specific use to others in the thalassemia field.

The thorough account is taken to extremes in reporting which commercial kits were the most sensitive and reproducible (lines 106-108). I recommend that this be either removed or at least put into context as "for the primers/reaction under study" and/or "in our hands".

Author Response

We thank Reviewer 4 for the positive evaluation and the time taken to assess our manuscript.

We agree with the Reviewer that without showing the full data for kit comparisons, which incidentally appears contraindicated given the current level of detail of the manuscript, we should not make blanket statements for the comparative performance of commercial products. At the same time, the selection of a robust kit proved critical for our own analyses, and we therefore did not remove reference to our initial comparison altogether, instead qualifying it in line with the Reviewer’s suggestion with the phrases “for the analyses in hand” and “for the current study” (lines 108-110).  

Round 2

Reviewer 3 Report

No further comments.